# Breakpoint modelling of temporal associations between non-pharmaceutical interventions and symptomatic COVID-19 incidence in the Republic of Ireland

**Martin Boudou**[1], **Coilin ÓhAiseadha**[1,2], **Patricia Garvey**[1,3], **Jean O'Dwyer**[1,4,5], **Paul Hynds**[1,5] *

**1** SpatioTemporal Environmental Epidemiology Research (STEER) Group, Environmental Sustainability & Health Institute, Technological University Dublin, Dublin, Ireland, **2** Health Service Executive, (HSE), Dublin, Ireland, **3** Health Protection Surveillance Centre (HPSC), Dublin, Ireland, **4** School of Biological, Earth and Environmental Sciences, University College Cork, Cork, Ireland, **5** Irish Centre for Research in Applied Geoscience, University College Dublin, Dublin, Ireland

* hyndsp@tcd.ie

**Data Availability Statement:** The data curator (Health Protection Surveillance Centre) and Research Ethics Review Committee have both

## Abstract

### Background

To constrain propagation and mitigate the burden of COVID-19, most countries initiated and continue to implement several non-pharmaceutical interventions (NPIs), including national and regional lockdowns. In the Republic of Ireland, the first national lockdown was decreed on 23$^{rd}$ of March 2020, followed by a succession of restriction increases and decreases (phases) over the following year. To date, the effects of these interventions remain unclear, and particularly within differing population subsets. The current study sought to assess the impact of individual NPI phases on COVID-19 transmission patterns within delineated population subgroups in the Republic of Ireland.

### Methods and findings

Confirmed, anonymised COVID-19 cases occurring between the 29$^{th}$ of February 2020 and 30$^{th}$ November 2020 (n = 72,654) were obtained. Segmented modelling via breakpoint regression with multiple turning points was employed to identify structural breaks across sub-populations, including primary/secondary infections, age deciles, urban/commuter/rural areas, patients with underlying health conditions, and socio-demographic profiles. These were subsequently compared with initiation dates of eight overarching NPI phases.

Five distinct breakpoints were identified. The first breakpoint, associated with a decrease in the daily COVID-19 incidence, was reported within 14 days of the first set of restrictions in mid-March 2020 for most population sub-groups. Results suggest that moderately strict NPIs were more effective than the strictest Phase 5 (National Lockdown). Divergences were observed across population sub-groups; lagged response times were observed among populations >80 years, residents of rural/ commuter regions, and cases associated with a below-median deprivation score.

imposed ethical/legal restrictions on sharing a de-identified dataset. As part of the the National Research Ethics Committee for COVID-19-related Health Research (NREC COVID-19) (Application number: 20-NRECCOV-061), the National Research Review Committee have specified that due to the inclusion of attributes relating to individual-level underlying health conditions and socioeconomic status, in concurrence with geo-referencing of cases at an extremely fine geographical resolution, individuals may potentially be identified, and as such, these data are not suitable for general availability. The National Research Ethics Review Committee may be contacted at nationaloffice@nrec.ie and will review all data acquisition applications on a case by case basis.

**Funding:** The presented study has been funded by the Science Foundation Ireland as part of its COVID-19 Rapid Response Funding Programme. PH received the research grant.The funders had no role in study design, data collection and analysis, decision to publish, or preparation of the manuscript

**Competing interests:** No authors have competing interests.

## Conclusions

Study findings suggest that many NPIs have been successful in decreasing COVID-19 incidence rates, however the strictest Phase 5 NPI was not. Moreover, NPIs were not equally successful across all sub-populations, with differing response times noted. Future strategies and interventions may need to be increasingly bespoke, based on sub-population profiles and required responses.

## 1. Introduction

Since its identification in late-2019 in Wuhan China, severe acute respiratory syndrome coronavirus 2 (SARS-CoV-2), the virus associated with coronavirus disease 2019 (COVID-19), has rapidly spread across the world [1]. The clinical presentation of infection by SARS-CoV-2 ranges between asymptomatic infection, mild symptomatic infection, and critical disease, defined by respiratory and/or multi-organ failure and death [1]. As of late March 2021, almost 127 million cases had been reported, resulting in approximately 2.8 million deaths [2] including 234,000 cases and 4,650 deaths in the Republic of Ireland (ROI) [3], representing unprecedented rates of hospitalisation and subsequent pressure on critical care services, both nationally and globally [4, 5]. The first laboratory confirmed-case reported in the ROI was reported on 29th February 2020, and within three weeks, cases had been confirmed in all 26 administrative counties [6]. On March 11th 2020, the World Health Organization declared COVID-19 a global pandemic, almost immediately after which a multi-faceted approach was adopted by the Irish government to reduce the impacts of the crisis and "flatten the (epidemic) curve". As no pharmaceutical intervention was available, this approach comprised an ensemble of non-pharmaceutical interventions (NPIs), a majority of which were rolled out nationally, with several regional NPIs (e.g., lockdowns) implemented later in 2020. Measures included 1.) limiting the spread of the virus in the community via school closures, closing the hospitality sector and social distancing 2.) contact tracing, 3.) ensuring adequate healthcare services and equipment available for those most impacted, and 4.) limiting the financial burden on the population, particularly business owners, arising from mitigation and containment measures [7]. A comprehensive overview of the key decisions and responses mandated by the Irish government are presented in Table 1.

In the early phases of the pandemic, a marked age-associated vulnerability in the burden of disease was established, with COVID-19-associated morbidity and mortality rates significantly higher among older sub-populations [8, 9]. More recently, studies have examined the impact of geographic location [10], and socioeconomic profile [11] on the likelihood of infection and subsequent outcomes (e.g., hospitalisation, severe infection, intensive care and mortality).

However, while there is little doubt as to the overarching efficacy of NPIs "flattening the curve" in the ROI and ensuring that healthcare systems remained intact, to date, it remains unclear how effectively individual measures (intervention phases) reduced viral transmission, and if measures were analogously efficacious among all population subsets and geographical regions. Accurate and reliable analyses of the epidemiology of COVID-19 as it relates to NPIs is essential for informing ongoing healthcare provision and future public health emergency planning. As such, the current study applied breakpoint linear regression analyses with multiple breakpoints to calculated daily incidence time-series for all symptomatic, laboratory-confirmed cases of COVID-19 among the Irish population from February 29th to November 30th 2020. Subsequently, identified structural breaks emanating from delineated sub-populations

**Table 1. Chronological summary of public health responses and non-pharmaceutical interventions (NPIs) implemented in the Republic of Ireland in response to COVID-19 Pandemic, March–October 2020 (Note: Due to ongoing national and regional/local changes to public health responses over the course of the study period, Table 1 is not a comprehensive description of all NPIs, but provides a summary of the most significant nationwide NPIs).**

| Date | Public Health Response | Restriction Increase/Decrease |
|------|------------------------|-------------------------------|
| 9th March 2020 | • St. Patrick's Day Festival Cancelled by Irish Taoiseach (Prime Minister) | Increase |
| 12th March 2020 | • Mandatory closure of schools, colleges, universities, childcare facilities, and state-run cultural institutions | Increase |
| | • Suspension of indoor gatherings for >100 people and outdoor gatherings for >500 people | |
| | • Workers urged to work from home, where possible | |
| 15th March 2020 | • Closure of pubs (bars) | Increase |
| 24th March 2020 | • Closure of non-essential businesses | Increase |
| | • All indoor and outdoor sporting activities cancelled | |
| | • All playgrounds/campgrounds closed | |
| | • Citizens not permitted to take unnecessary travel either within Ireland or overseas | |
| | • Physical distancing required when outside and social gatherings of no more than four individuals allowed (except for members of the same household) | |
| | • Citizens required to work from home unless they worked in essential services | |
| 27th March 2020 | • Stay at home measures announced for entire population (except essential workers) | Increase |
| | • Confinement radius of 2km from home address implemented | |
| | • No gatherings with anyone outside household | |
| | • People aged over 70 or medically vulnerable advised not to leave own home | |
| April 9th 2020 | • Irish Police Service granted legal powers to restrict movement, including arrest without warrant, under the Health Act 1947 (Section 31A-Temporary Restrictions) (Covid-19) Regulations). | Increase |
| May 18th 2020 | *Phase 1 of reopening of economy and society* | Decrease |
| | • Outdoor work and retail catering for outdoor work resumed | |
| | • Groups of up to four people are allowed to meet outdoors within 5 km of home. | |
| | • Outdoor public amenities, sport and fitness activities are allowed to open. | |
| June 8th 2020 | *Phase 2 of reopening of economy and society* | Decrease |
| | • Travel within a county or up to 20 km from home if crossing county borders is allowed. | |
| | • Groups of up to six people are allowed to meet either outdoors or indoors. | |
| | • Organised sporting, cultural or social activities for up to 15 people are allowed. | |
| | • Other retail (except within malls/shopping centres) are allowed to open. | |
| | • Funerals with up to 25 people in attendance are allowed. | |
| June 15th 2020 | Retail facilities in malls/shopping centres are allowed to open | Decrease |
| June 29th 2020 | *Phase 3 of reopening of economy and society* | Decrease |
| | • Domestic travel restrictions lifted. | |
| | • Cafes, restaurants, hotels, hostels, galleries, museums and pubs that serve food are allowed to open but social distancing must be maintained. | |
| | • Crèches reopen for essential workers and those who need childcare facilities | |
| | • Behind closed door sporting activities resumed. | |
| | • Higher risk retail outlets such as hairdressers are allowed to open. | |
| | • Indoor leisure facilities, festivals and cultural activities reopen. | |
| | • Indoor gatherings of up to 50 people and outdoor gatherings of up to 200 people allowed as long as public health advice followed. | |
| July 15th 2020 | • Face masks made mandatory in shops for customers and staff. | Increase |
| | • Maximum of 10 people from no more than 4 households allowed to visit other people's homes | |
| July 20th 2020 | • "Green list" of countries published; travellers from these countries can visit Ireland without having to quarantine. | Decrease |
| | • Advice to people living in Ireland is to avoid all non-essential overseas travel. | |

*(Continued)*

**Table 1.** (Continued)

| Date | Public Health Response | Restriction Increase/Decrease |
|---|---|---|
| August 10th 2020 | ***Phase 4 of reopening of economy and society:*** | **Decrease** |
| | • Crèches can reopen for the remaining workers. | |
| | • Weddings are permitted with limited attendance. | |
| | • Pubs/Nightclubs to remain closed | |
| August 18th 2020 | • Visitors to a home should be limited to not more than 6 from not more than 3 households | **Decrease** |
| | • Restaurants and Cafes to close by 11:30pm with a maximum of 6 per group (no more than 3 households) | |
| September 1st 2020 | Primary and Secondary Schools reopen | **Decrease** |
| October 7th 2020 | Restrictions levels increased to Level 3 (of a 5-point scale), including: | **Increase** |
| | • Visits to private homes limited to six people from two different households. | |
| | • Social family gatherings are suspended. | |
| | • Organized indoor gatherings are suspended while outdoor gatherings are limited to 15 people. | |
| | • Residents must remain in their counties of residence unless traveling for work, education, or other essential purposes. | |
| | • Public transport capacity is limited to 50 percent | |
| | • Restaurants and cafes allowed to remain open for takeaway and delivery | |
| October 21st 2020 | Six-week level 5 (most severe) lockdown, except for specific circumstances | **Increase** |
| | • 5km containment radius introduced | |
| | • Schools, early learning and childcare services remain open and are deemed essential | |
| | • Visits to other people's homes or gardens is banned | |
| | • Bars, cafes, restaurants and wet pubs may provide take-away and delivery services only. | |
| | • Public transport will operate at 25% capacity for the purposes of allowing those providing essential services to get to work | |
| | • Essential retail and services to remain open | |

Note: Restriction increase/decrease classification is based on the period immediately prior to new or adjusted interventions

(primary/secondary infection, age deciles, urban/commuter/rural, deprivation median, and patients with underlying health conditions) were compared with eight time-specific NPIs, accounting for the World Health Organisations (WHO) mean (5-day) and maximum (14-day) estimated COVID-19 incubation periods. We aimed to longitudinally estimate the efficacy, lag-period, and increasing/decreasing slope associated with specific NPIs among delineated sub-populations, with a view to providing governmental and public health authorities with a robust evidence-base for current, ongoing and future public health emergencies.

## 2. Methods

### 2.1 Case data

Anonymised notified COVID-19 case data were obtained from the Computerised Infectious Disease Reporting (CIDR) database (http://www.hpsc.ie/CIDR/), an information system used for the collation of notifiable (communicable) infection data in Ireland [12]. Address level data had already been geocoded to Small Areas by the Health Service Executive (HSE)-Health Intelligence Unit. COVID-19 incidence time-series were developed based on the "epidemiological date" (EpiDate) include in the HSE COVID-19 Case Surveillance Form.

### 2.2 Inclusion criteria

Due to the evolving testing policy since the start of the pandemic, only symptomatic cases were included for analyses. Accordingly, all laboratory confirmed cases, occurring between

29th February and 30th November 2020, with symptoms consistent with the Health Protection Surveillance Centre (HPSC) COVID-19 interim case definition (Version 6, January 27th 2021) [13] were included for anlyses. Accordingly, cases associated with detection of SARS-CoV-2 nucleic acid or antigen in a clinical specimen (Laboratory criteria), and exhibiting at least one of the following: sudden onset of cough or fever or shortness of breath or anosmia, ageusia or dysgeusia (clinical criteria, i.e. "symptomatic") were included.

## 2.3 Ethical considerations

Research ethical approval for use of the COVID-19 dataset and associated analyses were granted by the National Research Ethics Committee for COVID-19-related Health Research (NREC COVID-19) (Application number: 20-NREC-COV-061). All individual case data were fully anonymized before researcher acquisition.

## 2.4 Data subsetting

Sporadic (i.e., not recorded as associated with a confirmed outbreak or cluster) and outbreak index cases (the first case identified as part of a ecognized outbreak/cluster) were defined as primary cases, while all other known outbreak cases were defined as secondary cases.

Further, cluster incidence rate (based on CIDR outbreak code, with cluster initiation taken as the epidemiological date (EpiDate) attributable to second case within a defined cluster) per day was also defined and forwarded for analyses. Several metrics were additionally attributed to all individual clusters, including mean case age (years), mean duration (days) and mean size (case number) to investigate the effect of NPIs on cluster composition. All primary symptomatic cases of COVID-19 were discretized into decile-based age-groups for further analyses of age-based sub-populations and their responses to NPIs.

**2.4.1 Urban/rural classification.** A categorical SA-specific settlement type variable with three levels of classification was developed using data obtained from the Irish Central Statistics Office (CSO). The CSO settlement type dataset comprises six categories classified along an urban/peri-urban/rural scale ranging from 'city' (1) to 'highly rural/remote areas' (6). The classification variable was coded such that any classification which included a built-up area (classification 1 to 4) was recoded as 'urban', classification 5 (rural areas with high urban influence) was recoded as commuter/peri-urban, with all other areas (classification 6) coded as 'rural'.

**2.4.2 Deprivation index.** The Pobal Haase Pratschke (HP) Deprivation Index is derived from 16 individual components representing the three main dimensions of deprivation: demographic profile, social class composition, and labour market situation [14] (S1 Table). The relative index score represents a composite measure of deprivation based on these components, calculated for each CSO Small Area (SA) and measured on a single scale across all census periods (S2 Table), with the score acting as a comparative measure of deprivation between SAs during a census period [14]. Deprivation index data were obtained for 2016 (the most recent Irish census) to correspond with the study period, and binary classification used to delineate SAs based on high (above median relative score) and low (below median relative score) socioeconomic profile.

All datasets (urban/rural, deprivation) were spatially integrated using a unique SA identifier via the *match()* function (i.e., input vector), with subset-specific daily incidence rates calculated using the *ts()* function in R version 4.0.3 [15]

## 2.5 Non-pharmaceutical Interventions in the Republic of Ireland, March–November 2020

As shown (Table 1), several public health responses and non-pharmaceutical interventions (NPIs) were implemented across the Republic of Ireland during the study period, however, as

many of these were relatively minor adjustments and/or regionally specific, 8 primary time points were selected for comparison, based on the 5-phase COVID-19 Plan: Roadmap for Reopening Society and Business, published by the Irish Government in June 2020 (Table 2). As the purpose of the current study was qualifying the efficacy of nationwide NPIs, the aforementioned COVID-19 NPI phases have been employed for examination in the current study; the authors hypothesize that identification/quantification of minor NPI adjustments and/or regional NPI variations is significantly more complex, and not appropriate using the employed methodology (i.e., significantly lower regional incidence, etc.). It is important to note that, the COVID-19 NPI phases were not used to model time-series data; breakpoint modelling was undertaken entirely independently of the 8 time points presented in Table 2, with identified breakpoints compared with these dates following breakpoint identification.

## 2.6 Analytical methods

Segmented modelling via breakpoint regression is useful for assessing the effect of a covariate x (e.g., time-specific intervention) on the response y (e.g., incidence rate of infection), and has been widely used in medical and related research including mortality time-series [16], cancer incidence [17], and medication usage [18]. The "segmented" package [19] was used in the current study to fit several linear regression models between the response (laboratory-confirmed COVID-19 case number) and the explanatory variable (day number), thus allowing for identification of break-point estimates (i.e., dates on which COVID-19 incidence significantly increased/decreased). Identified breakpoints were subsequently compared with NPI phase dates using the World Health Organization's (WHO) mean (5-day) and maximum (14-day) estimated COVID-19 incubation periods, in order to ascertain the likely direct effect of specific NPI phase changes with notable shifts in the COVID-19 incidence time-series (i.e., if breakpoints were identified within 5 to 14 days following an NPI phase date, the authors believe this likely indicates a relatively direct cause-effect relationship). In the current study, multiple breakpoints were permitted for identification (based on multiple NPI phases and observed COVID-19 trends in the Republic of Ireland during the study period), with the breakpoint linear regression model thus defined as:

$$y_t = \beta_0 + \beta_1 x_t + \delta_1 (x_t - \tau_1)^+ + \cdots + \delta_k (x_t - \tau_k)^+, \ t = 1, 2, \ldots, n \qquad \text{(Eq 1)}$$

**Table 2. Non-pharmaceutical Intervention (NPI) phases in the Republic of Ireland, March–November 2020.**

| Date | Restrictions | Phase Equivalent |
|---|---|---|
| 15/03/2020 | Schools (12th) and Bars closed | 4 |
| 27/03/2020 | Stay At Home (SAH) Order–Full P5Lockdown | 5 |
| 18/05/2020 | Easing of COVID-19 P5 –Non-essential shops open–Outdoor sports and mixing with 4 people (max) permitted | 4 |
| 08/06/2020 | Easing of COVID-19 P4—Indoor and outdoor mixing up to 6 (max) people permitted | 3 |
| 29/06/2020 | Dry pubs/restaurants/barbers/indoor exercise permitted | 2 |
| 18/08/2020 | 15/6 people mixing restrictions outdoor/indoor | 2 |
| 06/10/2020 | P3 Restrictions implemented nationwide | 3 |
| 21/10/2020 | P5 Lockdown—Six weeks | 5 |

where $y_t$ is daily infection incidence (i.e., number of confirmed infections) modelled as a linear function of the explanatory variable, $x_t$, which is an ordinal number designating a day between 1 and 276 for the full time series, with $\tau_s$ (s = 1,2,.., k) representing identified breakpoints. In this case, k breakpoints divide the time into (k + 1) intervals (or sections), with β representing calculated interval gradients i.e., $\beta_1$ (first slope, $x_t < \tau_1$), $\beta_2 = \beta_1 + \delta_1$ (second slope, i.e., first slope plus difference in slopes), etc. An iterative approach was taken to determine the optimum number of breakpoints, whereby the npsi parameter (i.e., number of breakpoints) was increased from 1 up to a maximum of 10, using the adjusted $R^2$ value to optimise model fit without compromising model parsimony, as described by [19]. In summary, for the current study a "perfect fit" would be provided by $(x_{276} - 1)$ (i.e., 1 breakpoint per day within the time series, equating to the original time series). Thus, an increasing iterative approach was used to identify the lowest number of breakpoints required to explain the highest proportion of time-series variance and permit comparisons across sub-populations, without identifying identification of insignificant breakpoints (i.e., low gradient intervals). The authors hypothesize that identified breakpoints suggest abrupt changes in daily incidence (in addition to the date associated with this change), thus serving to retrospectively indicate which NPI phases were likely (in)effective based on a comparison of the breakpoint dates with NPI phase dates. Specifically, regarding those NPIs implemented shortly (≤14 days based on WHO maximum COVID-19 incubation period) before an abrupt decrease in slope as likely to have been effective, while those that were not followed by a decrease in slope within 14 days may be considered unlikely to have been effective.

## 3. Results

For individual (i.e., case-by-case) breakpoint analyses, 47,928 symptomatic COVID-19 cases (65.9% of all notified cases; 25,651 female (53.5%); mean age 41.2 years) were included (Fig 1), of which 61.5% (n = 29,459) of cases were classified as primary, and 18,469 (38.5%) classified as secondary cases. For breakpoint analyses of cluster number per day, the entire dataset including both symptomatic and asymptomatic cases (N = 72,311) was employed, based on the first reported epi-date associated with each cluster. Approximately 98.6% of symptomatic cases (n = 47,265) were successfully geocoded to one CSO SA, of which 15.3% (n = 7339), 70.8% (n = 33,950) and 12.5% (n = 5976) of cases were assigned to rural, urban and commuter/peri-urban categories, respectively; 1231 cases (2.6%) were associated with international

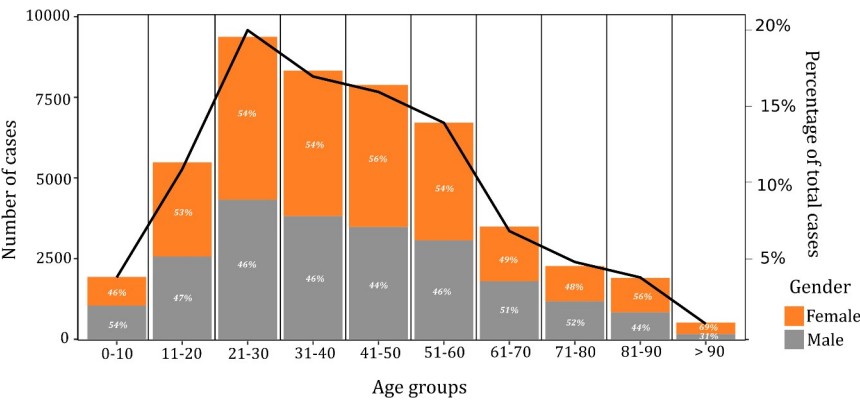

**Fig 1. Age and gender distribution of symptomatic COVID-19 cases in the Republic of Ireland, February 29th–November 30th 2020 (crude cases and percentage).**

travel. A median relative deprivation score of 0.870 (Minimum -36.18, Maximum 40.47; 25th and 75th percentiles: -6.42, 7.59) was used for case designation.

## 3.1 Breakpoint models

Case subsets were modelled to identify the best psi (breakpoint number) for use in breakpoint analyses (Table 3). A high degree of fit (minimum adjusted $R^2$ = 0.833) was found among all modelled subsets using psi = 5; while psi = 9 was found to result in higher fit values for several subsets (n = 7), findings of psi = 5 for all subsets are presented to increase comparability and clarity.

As shown (Table 3, Fig 2), while both primary and secondary case time-series achieved high degrees of fit via breakpoint modelling with psi = 5 (both $R^2$ >0.9), differing patterns were identified. The first breakpoint identified within the primary case time-series (29/03) occurred within 5 days of NPI Phase 4 (Fig 2A), while this occurred markedly later within the secondary case time series (23/04), and not within 14 days of the introduction of any NPI (Fig 2B). The first three breakpoints associated with secondary cases occurred over a significantly shorter time-period than observed within primary cases; a marked increase in secondary cases occurred within 14 days (02/09) of a relaxing of restrictions in mid-August (18/08) 2020, with this increase occurring approximately 3 weeks earlier among primary cases (07/08) (Fig 3). Both primary and secondary incidence rates exhibited a marked decrease within 5 days of the move from NPI Phase 2 to Phase 3 being implemented nationwide on 6th October; no break-point followed by a marked and consistent case decrease was identified within 5 or 14 days of either nationwide Phase 5 lockdowns.

The breakpoint model associated with cluster number per day (Fig 4) was relatively similar with respect to breakpoint location and interval gradient as that developed for primary cases. For example, the first identified breakpoint (18/03) leading to a consistent decline occurred

**Table 3. Case subsets, associated sample number and results of breakpoint modelling (based on 5 breakpoints), N = 47,928.**

| Case Subset | Number of cases in subset | Adj. $R^2$ |
|---|---|---|
| Primary | 29,459 | 0.925 |
| Secondary | 18,469 | 0.919 |
| Notified Case Clusters | 8581 | 0.923 |
| Underlying Health Conditions | 15,079 | 0.895 |
| 0–10 Years | 1933 | 0.883 |
| 11–20 Years | 5488 | 0.882 |
| 21–30 Years | 9380 | 0.889 |
| 31–40 Years | 8332 | 0.916 |
| 41–50 Years | 7883 | 0.904 |
| 51–60 Years | 6714 | 0.91 |
| 61–70 Years | 3494 | 0.87 |
| 71–80 Years | 2270 | 0.84 |
| 81–90 Years | 1907 | 0.889 |
| >90 Years | 518 | 0.833 |
| Rural | 7339 | 0.921 |
| Urban | 33,950 | 0.933 |
| Mixed/Commuter | 5976 | 0.848 |
| High Deprivation | 23,624 | 0.93 |
| Low Deprivation | 23,641 | 0.923 |

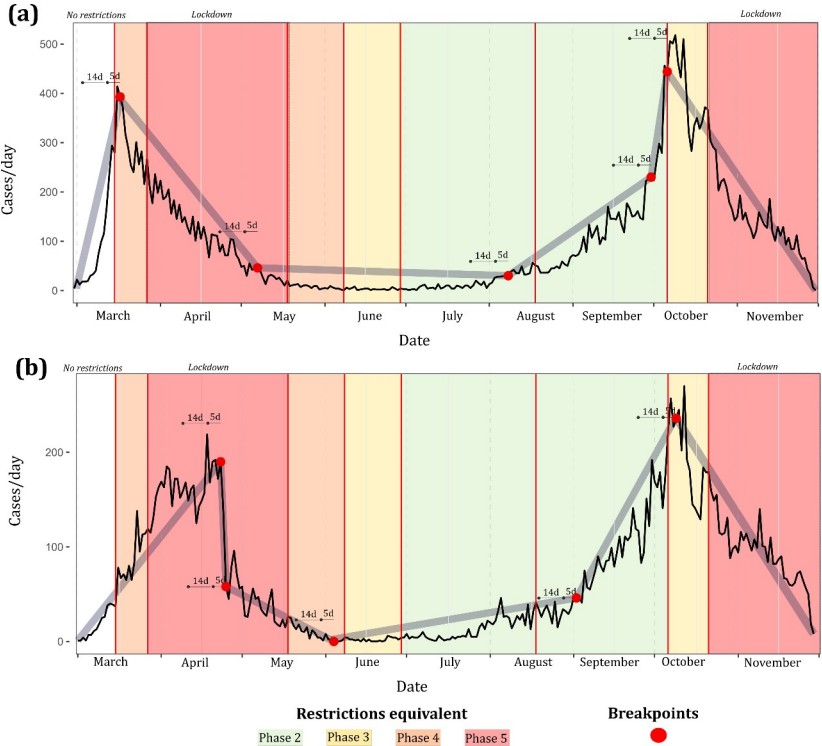

**Fig 2.** Breakpoint models (psi = 5) for a.) primary symptomatic COVID-19 and b.) secondary symptomatic COVID-19 in the Republic of Ireland (cases/day), February 29[th] to November 30[th] 2020.

within 5 days of initiation of Phase 4 closures, with another similar breakpoint identified one day before Phase 4 re-entry on October 6[th]. Again, no breakpoint followed by a decline in cluster occurrence were identified during or within 14 days of Phase 5 lockdowns. As shown (Table 4), no discernible pattern was observed with respect to breakpoint/interval order and cluster number, however, a notable monotonic decline in median within-cluster age (e.g., Interval 1 –Median Age 47.3 years; Interval 6 –Median Age 32.5 years) and cluster size (e.g., Interval 1 –Mean Cases/Cluster 10.3, Interval 6 –Mean Cases/Cluster 3.6).

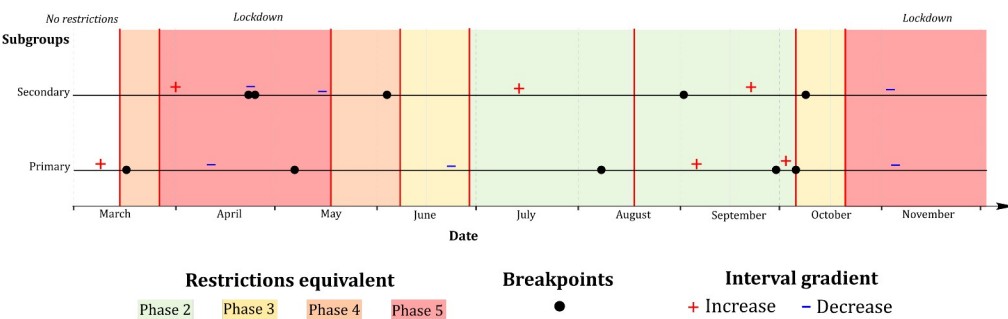

**Fig 3. Grid synthesis of primary and secondary time-series breakpoint models (psi = 5) in the Republic of Ireland (cases/day), February 29[th] to November 30[th] 2020; + and - signs refer to positive and negative interval gradients, respectively.**

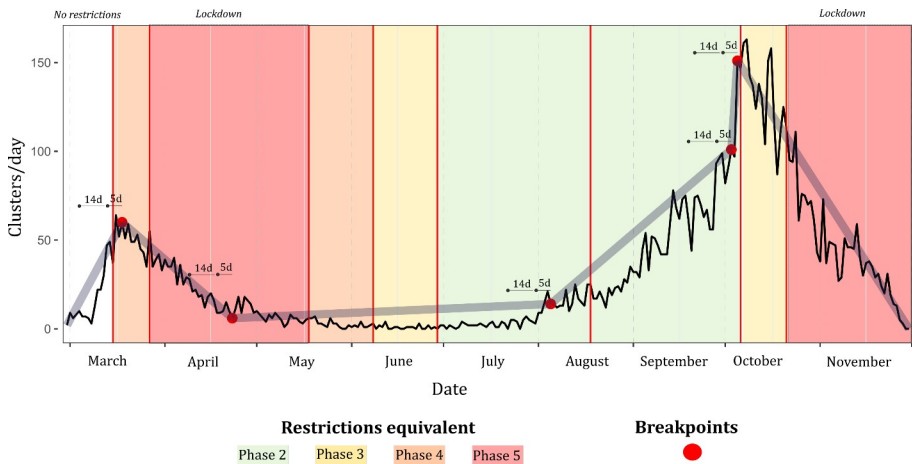

**Fig 4. Breakpoint model (psi = 5) for notified COVID-19 clusters in the Republic of Ireland (based on first reported epi-date), February 29th to November 30th 2020.**

Symptomatic COVID-19 cases among individuals with underlying health conditions based on the HSE COVID-19 Case Surveillance Form (chronic heart disease, hypertension, chronic neurological disease, chronic respiratory disease, chronic kidney disease, chronic liver disease, asthma requiring medication, immunodeficiency (including HIV), diabetes, BMI ≥40, cancer/malignancy) exhibited a distinctive breakpoint pattern (Fig 5). As for primary cases, secondary cases, and cluster number, the first identified breakpoint occurred relatively close to the 15/03 Phase 4 NPI, albeit not within the 5-day median incubation period (23/03). However, "relaxing" of NPIs from Phase 3 to 2 (29/06; 18/08) both coincided (within 14-days) with breakpoints followed by increasing daily incidence rates. The second wave peak and overarching pattern associated with this population subset was markedly different from that observed among other sub-populations.

All age deciles from 0–10 up to 71–80 years inclusive exhibited a first breakpoint occurring within the NPI Phase 4 (15/03 to 27/03), with an age-ordered pattern observed (i.e., younger deciles exhibited breakpoint before older deciles) (Fig 6A). The two oldest subsets, namely 81–90 years and >90 years, did not adhere to this pattern however, with an approximate 25-days gap between the first identified breakpoint among both these deciles and the 71–80-year decile. Both the 81–90 years and >90 years sub-populations exhibited markedly different breakpoint patterns over the duration of the study period. Conversely, while an age-ordered pattern was also identified for 2nd breakpoints, higher age deciles typically exhibited turning points prior to younger counterparts. For example, among 61–70-year-olds, a second breakpoint was

**Table 4. Results of breakpoint modelling for COVID-19 clusters per day, with associated median within-cluster age and mean cluster size.**

| Breakpoint Model Section | Start Date | End Date | Cluster Number | Median within-cluster age | Mean Cases/Cluster | Phase Duration (days) |
|---|---|---|---|---|---|---|
| 1 | 29/02/2020 | 18/03/2020 | 453 | 47.33 | 10.34 | 18 |
| 2 | 18/03/2020 | 23/04/2020 | 1085 | 45.00 | 7.07 | 36 |
| 3 | 23/04/2020 | 05/08/2020 | 421 | 37.33 | 5.20 | 104 |
| 4 | 05/08/2020 | 03/10/2020 | 2546 | 34.00 | 4.18 | 59 |
| 5 | 03/10/2020 | 05/10/2020 | 248 | 33.83 | 4.07 | 2 |
| 6 | 05/10/2020 | 30/11/2020 | 3828 | 32.50 | 3.57 | 56 |

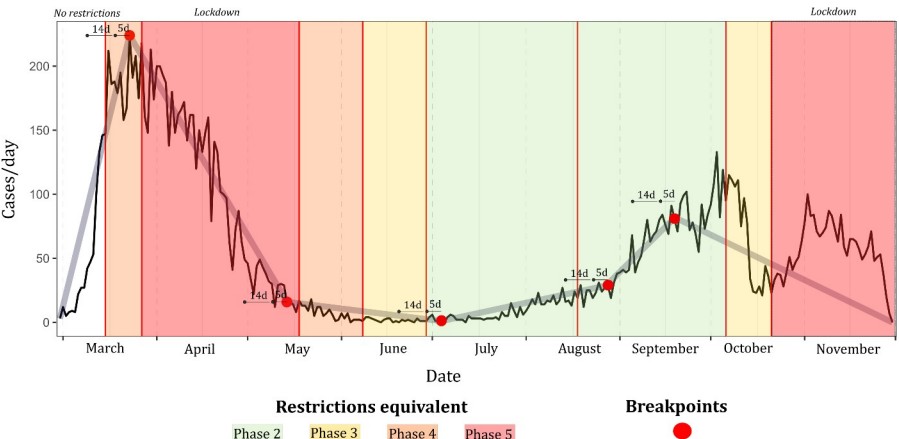

**Fig 5. Breakpoint model (psi = 5) for symptomatic COVID-19 in the Republic of Ireland among persons with underlying health conditions (cases/day), February 29th to November 30th 2020.**

identified 10 days (08/05) before easing of the first Phase 5 lockdown (18/05), while this breakpoint occurred one week after easing among the 31–40-year decile (23/05). All age-based sub-populations exhibited a breakpoint followed by a negative interval gradient (Range -0.18 - -2.8) within 14 days of nationwide Phase 3 restrictions (06/10), with no breakpoints identified during the ensuing Phase 5 lockdown (21/10 onwards).

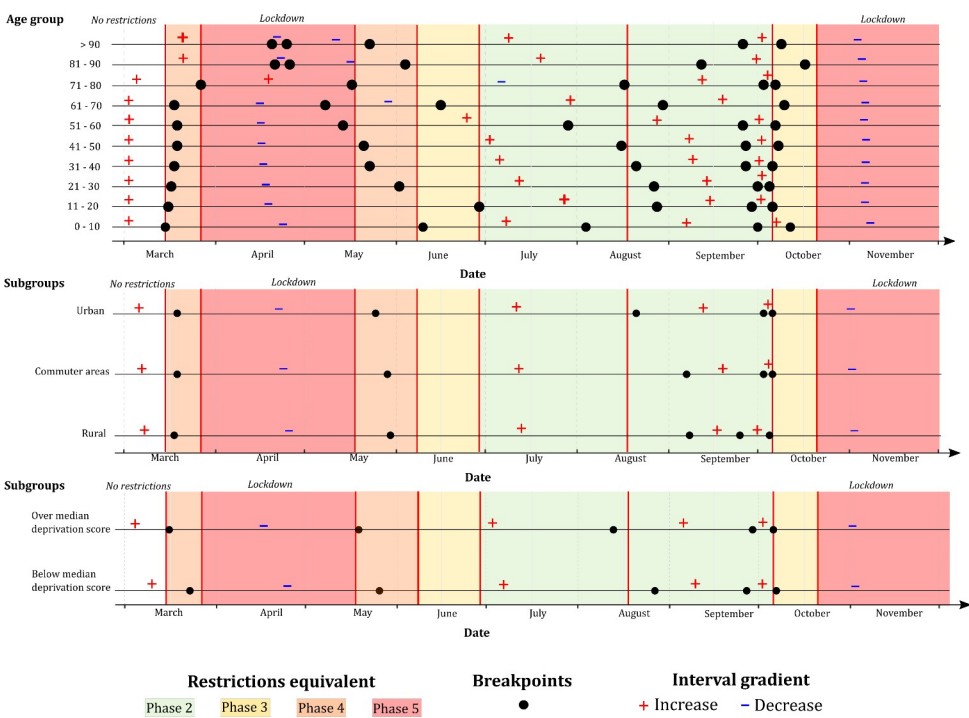

**Fig 6. Grid synthesis of (top) age-related deciles, (middle) urban/commuter/rural classification, and (bottom) above/below median deprivation score time-series breakpoint models (psi = 5) in the Republic of Ireland (cases/day), February 29th to November 30th 2020; + and - signs refer to positive and negative interval gradients, respectively.**

A similar pattern was observed between identified breakpoints based on urban/rural classification during the first half of the study period (February-August) (Fig 6B), for example, while the second identified breakpoint among urban cases occurred slightly earlier (25/05) than among cases residing in rural (30/05) or commuter areas (29/05), these differences were relatively minor, with all three breakpoints, followed by low positive gradient intervals (Range 0.16–0.57), occurring within 14 days of Phase 5 easing (18/05). The most significant divergence from this pattern was represented by the third identified breakpoint, all of which were followed by positive gradient intervals (1.35–5.05), which occurred on 21/08 within the urban case time-series, and approximately two weeks later in rural (08/09) and commuter areas (07/09).

As shown (Fig 6C), breakpoints identified within the sub-population associated with the "above-median" (i.e., low) deprivation score (> 0.87) typically occurred prior to or in concurrence with those associated with a deprivation score <0.87. For example, the first identified breakpoint within the low deprivation sub-population occurred on 16/03 (one day after P4 initiation), while the first high-deprivation breakpoint occurred one week later (23/03). Similarly, the second low-deprivation breakpoint occurred one day after Phase 5 easing (19/05), while this breakpoint occurred one week later among those residing in high-deprivation SAs (26/05), thus indicating that NPI responses occurred more rapidly within low-deprivation areas in the early phases of the pandemic, with this pattern dissipating over time.

## 4. Discussion

Results from the current study may offer valuable insight into the mechanisms of NPIs and their efficacy across sub-populations, geographic regions and sociodemographic profiles. Five distinct breakpoints were identified and compared across the study period, which consisted of eight varying (increasing/decreasing stringency) stages of non-pharmaceutical intervention. The initial identified breakpoint among the majority of case subsets (excluding secondary infection, and infection among those >80 years) followed by a significantly negative gradient, occurred within 5–7 days of the first set of restrictions in mid-March 2020, indicating extensive 'voluntary' societal change. This is likely reflective of the socially perceived 'high risk' at the time, due to unprecedented media exposure and scale of the unfolding event, resulting in high compliance (e.g., handwashing, social distancing, staying at home, etc.). This effect has been previously reported; restaurant reservations in the United States and movie revenues in Sweden were shown to reduce significantly before any imposition of NPIs, not as a direct response [20]. Thus, a large proportion of decreased mobility in both countries during the initial stages of the pandemic was voluntary, driven by the number of COVID-19 cases, likely proxying for greater awareness of risk, with the total contribution of NPIs moderate compared to voluntary actions [20].

All identified second breakpoints occurred between mid-April and mid-May, after which COVID-19 incidence rates remained consistently low for 2–3 months, despite eased restrictions (i.e., Phase 2). Previous studies have shown that respiratory viruses, including coronaviruses, exhibit significantly lower incidence rates during summer, and particularly in temperate regions like Ireland, with colder conditions during winter a major driver for respiratory tract infections due to increased virus stability and transmission and a weakened host immune system [21]. Additionally, Merow & Urban (2020) posit that increased UV radiation (viral inactivation) in conjunction with increased vitamin D production (via increased UV radiation) may play a key role, particularly when combined with ongoing voluntary social precautions; models suggest that up to 36% of the variation in maximum COVID-19 growth rates in the US was attributed to short-term weather, with UV light the most strongly associated climatic variable with lower COVID-19 growth rates [22].

The fourth identified breakpoint within all subsets marked a significant and abrupt increase of infection rates from mid to late September 2020, likely attributable to the reopening of schools (both primary and secondary) and the return of college/university students to campus accommodation. While research from Ireland has concluded that there is no evidence of secondary transmission of COVID-19 from children attending school [23], the reopening of schools likely resulted in increased social interaction between parents/caregivers. Moreover, Santamaria & Hortal (2020) suggest that numerous pandemic waves and consequent NPI phases may lead to lapsing vigilance associated with fatigue or lowered risk perception [24]. In response to the increasing number of notifications in line with breakpoint four, the final breakpoint (significant negative interval gradient across all subsets) occurred simultaneously with the move to Phase four of the national response plan, again indicating a natural reduction in disease prevalence prior to a mandated NPI, perhaps resulting from an increased social perception of risk and reinforced personal protective measures. The re-introduction of increasingly restrictive NPIs immediately after this period was arguably unnecessary; notably no significant breakpoint followed by a negative incidence gradient were identified during either of the 'high-level' (i.e, Phase 5) NPI phases, as 'voluntary' measures were likely already triggered (i.e., increased media attention, etc.). While several mathematical modelling studies and meta-analyses have reported a marked reduction in COVID-19 incidence and mortality [25–27], a meta-analysis of 87 regions including 3741 pairwise comparisons could not explain if COVID-19 mortality is reduced by "stay at home" orders in 98% of comparisons [28]. Analysis of incidence trends in Germany detected a crucial breakpoint on 8 March, coinciding with the cancellation of mass events on the same day, with the authors suggesting that increased awareness and voluntary changes in behaviour, e.g., social distancing, respiratory etiquette and hand hygiene, likely had a significant effect [29]. Similarly, breakpoint analysis of daily incidence and effective reproductive number in Spain indicates reductions in disease growth preceding mobility restrictions [24], with Post et al. (2021) also reporting that, although breakpoints in daily effective contact rates (ECR) aligned with government interventions, ECRs after full lockdown were not necessarily lower than that after a ban on gatherings alone [30]. As such, it appears that the efficacy of Phase 5 lockdowns may be regionally- or nationally-specific, and in the case of Ireland, there is no evidence to suggest that Phase 5 NPIs were more effective than Phase 3/4 NPIs during the first two "waves" of infection. Considering the potential impacts of prolonged restrictions on psychological wellbeing [31, 32], significant further work and a robust evidence base is required to support future large-scale use of these measures.

Breakpoints identified among both secondary cases and recognised COVID-19 clusters differed from patterns observed among primary cases (Figs 3 & 4); within the cluster number subset, breakpoints preceded lockdown by a week or more in both first and second wave, while the crucial breakpoint identified for secondary cases during the first wave occurred three weeks after Phase 5 lockdown was imposed. This may indicate that measures required to control outbreaks may differ from those required to control primary transmission within the general population. Likewise, when cases were assessed based on age profile (Fig 6), marked variability was observed; a notable decrease in "quick" response was noted among individuals >80 years i.e., a reduction in the number of cases occurred significantly more quickly among the population <80 years. COVID-19 incidence rate breakpoints were notably staggered based on age decile; case incidence decreases occurred in younger populations first (approximately 1–3 days per decile), and case increases occurring more quickly among older individuals (approximately 2–5 days per decile). This could be attributable to numerous factors including a longer incubation period among older adults [9], comorbidity-related infection severity and reduced immune response [33], all of which may cause a lengthened persistence of disease within elderly populations.

The history of the pandemic indicates that nursing homes, or 'long-term care facilities' (LTCF) for older people, in Ireland as in other countries, were severely impacted during the first wave [3], which required specific outbreak control measures to break the chain of transmission. As these infections occurred within the LTCF setting, they were not amenable to the effects of NPI's implemented within the general population.

The timing of the third identified breakpoint differed with respect to settlement type, occurring in mid-August in the urban subgroup and not until approximately 7th September in the rural subgroup (Fig 6B). This may reflect differing social activities, e.g., an earlier seasonal increase in indoor gatherings among urban versus rural residents and/or condensed residences in urban areas expediting SAR-CoV-2 transmission. A recent study from South Carolina reports that relaxing NPIs was followed by infection hotspots reappearing in urban areas more rapidly than rural areas [34], with the authors recommending locally- or regionally-customised interventions. Henning-Smith et al. (2020) suggest that rural residents are typically older, more likely associated with underlying health conditions, and less likely to have access to healthcare and/or necessary financial resources [35]. As such, a "one size fits all" approach to NPIs may not be appropriate based upon geographic location, which in Ireland has been associated with numerous potential drivers of infection including occupation, income, car ownership, educational attainment, and level of affluence [36]. It is important to note that, a significant association was found between rurality and deprivation within the Irish COVID-19 dataset; rural cases were associated with a median HP Deprivation Score of -8.25, while urban cases exhibited a median deprivation score of -3.22, and as such, it is difficult to make recommendations solely based on geographic location. Similarly, a relationship exists between deprivation and underlying health (based on underlying health condition number associated with confirmed cases) in Ireland, adding further complexity.

The earlier first breakpoint identified among cases associated with a higher deprivation score (i.e., increased affluence) may indicate greater awareness and willingness to adopt voluntary measures for self-protection, while working from home (and thus complying with stay-at-home orders) may be more difficult for those in lower paid jobs. Conversely, the earlier timing of second and third breakpoints may point to higher levels of susceptibility among more deprived sub-populations, based on underlying health status, as mentioned above or increased household density. A retrospective cohort study comprising 3528 patients with laboratory confirmed COVID-19 residing in New York (US) found that patients associated with high poverty areas were significantly younger, exhibited higher prevalence of comorbidity and were more likely to be female or from a racial minority compared to individuals living in low poverty areas [37].

The presented study is the first of its kind from Ireland, insofar as it the first to employ temporally-specific case data, geo-coded to a very fine resolution, thus allowing delineation of cohorts pertaining to geographical attributes including deprivation and urban/rural classification. Notwithstanding, there are some inherent limitations that should be considered. The epi-date used to develop sub-population time-series, while complete, included >1 temporal classification; while a large majority of epi-dates were classified as "infection onset date", other case epi-dates were classified as "lab specified collection date", "event creation date", and "date of diagnosis". Accordingly, presented findings should be interpreted with an appropriate level of caution. Similarly, it was not possible to accurately delineate large LTCH or workplace outbreaks, which may influence, and potentially bias, the number of cases attributed to specific population subsets e.g., urban/commuter/rural classification. Lastly, COVID-19 surveillance data may be biased due to geographical distribution of testing locations, and particularly as they pertain to secondary and/or asymptomatic cases. Consequently, it is important to note that cluster number, median within-cluster age and cluster size (i.e, notified cases per cluster)

were inherently influenced by the availability of testing for asymptomatic individuals, and particularly during the early phases of the pandemic.

## Supporting information

**S1 Table. Pobal HP deprivation index components and descriptions.**
(DOCX)

**S2 Table. Pobal HP Index absolute and relative deprivation index score classification.**
(DOCX)

## Author Contributions

**Conceptualization:** Martin Boudou, Paul Hynds.

**Data curation:** Coilin ÓhAiseadha, Patricia Garvey, Paul Hynds.

**Formal analysis:** Martin Boudou.

**Funding acquisition:** Jean O'Dwyer, Paul Hynds.

**Investigation:** Martin Boudou, Jean O'Dwyer, Paul Hynds.

**Methodology:** Jean O'Dwyer, Paul Hynds.

**Project administration:** Paul Hynds.

**Resources:** Coilin ÓhAiseadha, Patricia Garvey.

**Supervision:** Jean O'Dwyer, Paul Hynds.

**Validation:** Martin Boudou, Paul Hynds.

**Visualization:** Martin Boudou.

**Writing – original draft:** Martin Boudou, Jean O'Dwyer, Paul Hynds.

**Writing – review & editing:** Coilin ÓhAiseadha, Patricia Garvey.

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
