## [Decision Letter · Decision Letter 0]

9 Jun 2021

PONE-D-21-10957

Breakpoint modelling of temporal associations between non-pharmaceutical interventions and the incidence of symptomatic COVID-19 in the Republic of Ireland

PLOS ONE

Dear Dr. Hynds,

Thank you for submitting your manuscript to PLOS ONE. After careful consideration, we feel that it has merit but does not fully meet PLOS ONE’s publication criteria as it currently stands. Therefore, we invite you to submit a revised version of the manuscript that addresses the points raised during the review process.

We look forward to receiving your revised manuscript.

Kind regards,

Lucy C. Okell

Academic Editor

PLOS ONE

Journal Requirements:

2. Thank you for stating in the text of your manuscript "Research ethical approval for use of the COVID-19 dataset and associated analyses were granted by the National Research Ethics Committee for COVID-19-related Health Research (NREC COVID-19) (Application number: 20-NREC-COV-061)". Please also add this information to your ethics statement in the online submission form.

3. In your ethics statement in the Methods section and in the online submission form, please provide confirm that all data were fully anonymized before you accessed them.

5. Please amend the manuscript submission data (via Edit Submission) to include author Garvey, P.

Reviewers' comments:

Reviewer's Responses to Questions

**Comments to the Author**

1. Is the manuscript technically sound, and do the data support the conclusions?

Reviewer #1: Yes

Reviewer #2: Partly

2. Has the statistical analysis been performed appropriately and rigorously? 

Reviewer #1: Yes

Reviewer #2: I Don't Know

3. Have the authors made all data underlying the findings in their manuscript fully available?

Reviewer #1: No

Reviewer #2: No

4. Is the manuscript presented in an intelligible fashion and written in standard English?

Reviewer #1: Yes

Reviewer #2: Yes

5. Review Comments to the Author

Reviewer #1: 1) Formula (1) was not clearly explained. I suggest removing it as the current formula (2) is sufficinet. Please properly using subscript in formula (2).

2) In Abstract, line 33, it should be “NPIs” instead of “NCIs”.

Reviewer #2: It is important to know the effects of specific interventions during the current COVID-19 outbreak. The study purpose is to estimate the efficacy, lag period, and increasing/decreasing slope associated with specific NPIs among delineated sub Populations in Ireland. The topic is interesting. Boudou et al has collected case data and perform regression analysis with different breakpoints. The regression results are related to different intervention time point. However, In the current draft, the efficacy and lag period associated with specific NPIs are not described well. The purpose of using this breakpoint regression is not clear.

My concerns are here.

1. Authors reduced 16 NPI periods to 8 time points. What is the rational of this reduction? Mandatory face mask wearing in shops, which is implemented on July 15 is removed. Also, are there other face mask regulations? Such as face mask wearing on public transport? I found that there is such rule, but not described in Table 1.

2. Authors applied breakpoint linear regression analyses with multiple breakpoints. The methods are quite different than most infectious disease modelling works. I admit I am not familiar with breakpoint linear regression. Maybe because this is not a common approach in regression analysis, I found that it is difficult to understand how this approach was used in the Methods. E.g., in 2.3 Analytical Methods, how breakpoints were identified was not explained. In the equation (2) in Line 186, dependent variable Y is not explained. δ is not explained.

3. Please discuss why use breakpoints less the total number of intervention time points? Why not just analyse the slope during each of the intervention interval and obtain the slope. These allow us to understand the benefit and the limitation of this approach.

4. Cannot understand the approach how number and the value (location) of breakpoint are calculated. In line 191, “An iterative approach was taken to optimise breakpoint number via increasing the npsi parameter from 1”. What is npsi? The sentence “An iterative approach was taken to optimise breakpoint number via increasing the npsi parameter from 191 1, up to a maximum of 10, using the adjusted R2 value to balance acceptable model fit with model parsimony, for ease of interpretation, thus providing a number (npsi) of optimised breakpoints with a sequence date (from 1 to 276, i.e., length of the time series), matched to the corresponding date and associated (n + 1) linear intervals and corresponding interval slope.” is too long and too complicated to understand. What does it mean “using the adjusted R2 value to balance acceptable model fit with model parsimony”. I presumed that most of the regression model is based on likelihood. Maybe this approach is different, but I cannot understand the approach.

5. Line 188, xt is “day number”. Please describe what exact xt is in terms of the outbreak. If this only refers to symptomatic cases, why in Line 203, it was described “the entire dataset (N = 72,311, i.e., both symptomatic and asymptomatic) was included for breakpoint analyses…)”.

6. I don’t understand why look breakpoints of primary and secondary cases separately as shown in Figure 2. Why not look the sum of primary and secondary cases separately?

7. The description in Table 1 and 2 are unclear. For example, why on Aug 18, Restaurants and Cafes to close by 11:30pm with a maximum of 6 per group (no more than 3 households) represent a decrease in restriction? Comparing to which one? Because on June 29 Cafes, restaurants, hotels, hostels, galleries, museums and pubs that serve food are allowed to open but social distancing must be maintained.

8. In the current draft, authors is not describing well the efficacy and lag period associated with specific NPIs. Where should we find the efficacy and the lag? Should the change in slope be interpreted as the efficacy? Why not estimate the growth rate in each interval? I will suggest to list all the effects of NPIs in the column of Table 2?

9. My suggestion is, since the breakpoint can be located different than the time NPI was implemented, the benefit of using these breakpoint should be clearly stated. In seems like breakpoints can be associated with deprivation score (In Line 318, ‘breakpoints identified within the sub-population associated with the “above-median”’). This may be a good reason. However, I am not so clear why these two are associated. Break points represent different time points. Please explain clearly how the association was determined.

10. The discussion is too long, with 4 pages. Please make them concise to contain most important messages.

6. PLOS authors have the option to publish the peer review history of their article (what does this mean?). If published, this will include your full peer review and any attached files.

Reviewer #1: No

Reviewer #2: No

---

## [Author Response · Author response to Decision Letter 0]

29 Jun 2021

PONE-D-21-10957

Breakpoint modelling of temporal associations between non-pharmaceutical interventions and the incidence of symptomatic COVID-19 in the Republic of Ireland

Response to Editor and Reviewers

Editor/Journal Comment 1: 

Author Response 1:

We have checked (and double-checked) this. All files now comply with the journals style requirements and file naming formats

Editor/Journal Comment 2: 

Thank you for stating in the text of your manuscript "Research ethical approval for use of the COVID-19 dataset and associated analyses were granted by the National Research Ethics Committee for COVID-19-related Health Research (NREC COVID-19) (Application number: 20-NREC-COV-061)". Please also add this information to your ethics statement in the online submission form.

Author Response 2:

Amended as requested

Editor/Journal Comment 3: 

In your ethics statement in the Methods section and in the online submission form, please provide confirm that all data were fully anonymized before you accessed them.

Author Response 3:

Amended as requested

Editor/Journal Comment 4: 

We note that you have indicated that data from this study are available upon request. PLOS only allows data to be available upon request if there are legal or ethical restrictions on sharing data publicly. For information on unacceptable data access restrictions, please see http://journals.plos.org/plosone/s/data-availability#loc-unacceptable-data-access-restrictions.

Editor/Journal Comment 5: 

Please amend the manuscript submission data (via Edit Submission) to include author Garvey, P.

Author Response 5:

Amended as requested (Apologies, a genuine error!)

Editor/Journal Comment 6: 

Please include captions for your Supporting Information files at the end of your manuscript, and update any in-text citations to match accordingly. Please see our Supporting Information guidelines for more information: http://journals.plos.org/plosone/s/supporting-information.

Author Response 6:

Amended as requested

 Reviewer #1: 

Reviewer Comment 1:

Formula (1) was not clearly explained. I suggest removing it as the current formula (2) is sufficinet. Please properly using subscript in formula (2).

Author Response 1:

Amended as requested 

Reviewer Comment 2:

In Abstract, line 33, it should be “NPIs” instead of “NCIs”.

Author Response 2:

Amended as requested 

Reviewer #2: 

It is important to know the effects of specific interventions during the current COVID-19 outbreak. The study purpose is to estimate the efficacy, lag period, and increasing/decreasing slope associated with specific NPIs among delineated sub Populations in Ireland. The topic is interesting. Boudou et al has collected case data and perform regression analysis with different breakpoints. 

Author: Thank you very much! We’re happy that you appreciate the importance of the work!!

Reviewer Comment 1. 

Authors reduced 16 NPI periods to 8 time points. What is the rational of this reduction? Mandatory face mask wearing in shops, which is implemented on July 15 is removed. Also, are there other face mask regulations? Such as face mask wearing on public transport? I found that there is such rule, but not described in Table 1.

Author Response 1:

The reviewer is quite right, insofar as, we have chosen to discretize the number of NPI periods for study. In fact, there were many more than 16 NPI periods over the course of the study periods, however many of these were characterise as being very small “tweaks” to existing NPIs or were local/regional in nature. Accordingly, we have chosen to amend the Table 1 title in order to make this clearer for readers (See below). We have also elected to “delete” mention of “16 NPI periods” as we believe this terminology is/was somewhat misleading, and thus Section 2.2 has now been amended to make this clearer also (See below). We believe that this is now far more transparent/understandable for readers, with the primary misunderstanding occurring due to our use of “16 NPI Periods”, which was not strictly accurate. Also, the lack of a caveat for Table 1 may have led the reader to believe that this Table/list was absolutely comprehensive

“Table 1. Chronological summary of public health responses and non-pharmaceutical interventions (NPIs) implemented in the Republic of Ireland in response to COVID-19 Pandemic, March – October 2020 (Note: Due to ongoing national and regional/local changes to public health responses over the course of the study period, Table 1 is not a comprehensive description of all NPIs, but provides a summary of the most significant nationwide NPIs)”

Section 2.2

“As shown (Table 1), several public health responses and non-pharmaceutical interventions (NPIs) were implemented across the Republic of Ireland during the study period, however, as many of these were relatively minor adjustments and/or regionally specific, 8 primary time points were selected for comparison, based on the 5-phase COVID-19 Plan: Roadmap for Reopening Society and Business, published by the Irish Government in June 2020 (Table 2). As the purpose of the current study was identification and, where possible, quantification of the efficacy of nationwide NPIs, the aforementioned 8-point, COVID-19 NPI phases have been employed for examination in the current study; the authors hypothesize that identification/quantification of minor NPI adjustments and/or regional NPI variations is significantly more complex, and not appropriate using the employed methodology (i.e., significantly lower regional incidence, etc.).”

Reviewer Comment 2. 

Authors applied breakpoint linear regression analyses with multiple breakpoints. The methods are quite different than most infectious disease modelling works. I admit I am not familiar with breakpoint linear regression. Maybe because this is not a common approach in regression analysis, I found that it is difficult to understand how this approach was used in the Methods. E.g., in 2.3 Analytical Methods, how breakpoints were identified was not explained. In the equation (2) in Line 186, dependent variable Y is not explained. δ is not explained.

Author Response 2: 

We agree, more detail and clarity were required here. Accordingly, we have significantly rewritten Section 2.3 (below), to make it easier to understand, in addition to inserting more detail to explain equation 2, and particularly with respect to yt and δ1. Please note, in response to Reviewer #1, we have removed formula/equation (1), so formula/equation (2) has been renumbered as (1).

2.3 Analytical Methods 

“Segmented modelling via breakpoint regression is useful for assessing the effect of a covariate x (e.g., time-specific intervention) on the response y (e.g., incidence rate of infection), and has been widely used in medical and related research including mortality time-series [16], cancer incidence [17], and medication usage [18]. The “segmented” package [19] was used in the current study to fit several linear regression models between the response (laboratory-confirmed COVID-19 case number) and the explanatory variable (day number), thus allowing for identification of break-point estimates (i.e., dates on which COVID-19 incidence significantly increased/decreased). Identified breakpoints were subsequently compared with NPI phase dates using the World Health Organization’s (WHO) mean (5-day) and maximum (14-day) estimated COVID-19 incubation periods, in order to ascertain the likely direct effect of specific NPI phase changes with notable shifts in the COVID-19 incidence time-series (i.e., if breakpoints were identified within 5 to 14 days following an NPI phase date, the authors believe this likely indicates a relatively direct cause-effect relationship). In the current study, multiple breakpoints were permitted for identification (based on multiple NPI phases and observed COVID-19 trends in the Republic of Ireland during the study period), with the breakpoint linear regression model thus defined as: 

 yt = β0 + β1xt + δ1 (xt − τ1 ) + + ⋯ + δk (xt − τk ) +, t = 1,2, … , n (Eqn. 1)

where yt is daily infection incidence (i.e. number of confirmed infections) modelled as a linear function of the explanatory variable, xt, which is an ordinal number designating a day between 1 and 276 for the full time series, with τs (s = 1,2, . . , k) representing identified breakpoints. In this case, k breakpoints divide the time into (k + 1) intervals (or sections), with β representing calculated interval gradients i.e., β1 (first slope, xt < τ1 ), β2 = β1 + δ1 (second slope, i.e., first slope plus difference in slopes), etc. An iterative approach was taken to determine the optimum number of breakpoints, whereby the npsi parameter (i.e., number of breakpoints) was increased from 1 up to a maximum of 10, using the adjusted R2 value to optimise model fit without compromising model parsimony, as described by Muggeo (2013). In summary, for the current study a “perfect fit” would be provided by (x276 − 1) (i.e., 1 breakpoint per day within the time series, equating to the original time series). Thus, an increasing iterative approach was used to identify the lowest number of breakpoints required to explain the highest proportion of time-series variance and permit comparisons across sub-populations, without identifying identification of insignificant breakpoints (i.e., low gradient intervals).” 

Reviewer Comment 3. 

Please discuss why use breakpoints less the total number of intervention time points? Why not just analyse the slope during each of the intervention interval and obtain the slope. These allow us to understand the benefit and the limitation of this approach

Author Response 3: 

Given that the pre-existing evidence base is poor (due to the absolutely unprecedented nature of the global situation), breakpoints were derived from analysis of the empirical data without any a priori assumption that the interventions are/were effective (or ineffective) i.e., we have no evidence on which to base any working hypotheses. Segmented regression is typically deployed where the overarching relationship between the response and some explanatory variable(s) is non-linear, exhibiting a pattern whereby the effect on the response changes abruptly i.e., breakpoints (similar to interrupted time-series, however interrupted time-series are typically employed to investigate a stand alone/individual intervention, not a series of intervention increases and decreases). In the present study, we set out to identify breakpoints and their potential for indicating if and which of the many NPIs were (most) effective and which events associated with the easing of restrictions may have triggered a renewed increase in incidence. In effect, breakpoint analysis tests the null hypothesis that the NPIs have no effect. The identification of breakpoints and the fact that some of these were found to coincide with NPIs in the absence of other confounders thus indicates the null hypothesis is false in some cases. Conversely, even though we detected breakpoints, we cannot assume that these are triggered by NPIs, since they might alternatively result from increasing population immunity, seasonal effects, or other influences on virus transmission. 

Overall, we considered it appropriate to analyse the empirical data without any preconceived hypotheses regarding the efficacy of interventions, and to consider the potential influence of NPIs retrospectively from our identification of the breakpoints. While we absolutely see the reviewers point that the slope between NPIs could be examined, this would not account for the lagged response to NPIs i.e., if we simply compare slopes between each NPI date, this presumes an “immediate response” to the NPIs, and does not account for differing lags based on behavioural changes, immunological responses, etc etc between different subsets. In order to ensure readers understand this point, we have added the following statement directly before Table 2 in the corrected manuscript: 

“It is important to note that, the COVID-19 NPI phases were not used to model time-series data; breakpoint modelling was undertaken entirely independently of the 8 time points presented in Table 2, with identified breakpoints compared with these dates following breakpoint identification.“

Reviewer Comment 4. 

Cannot understand the approach how number and the value (location) of breakpoint are calculated. In line 191, “An iterative approach was taken to optimise breakpoint number via increasing the npsi parameter from 1”. What is npsi? The sentence “An iterative approach was taken to optimise breakpoint number via increasing the npsi parameter from 191 1, up to a maximum of 10, using the adjusted R2 value to balance acceptable model fit with model parsimony, for ease of interpretation, thus providing a number (npsi) of optimised breakpoints with a sequence date (from 1 to 276, i.e., length of the time series), matched to the corresponding date and associated (n + 1) linear intervals and corresponding interval slope.” is too long and too complicated to understand. What does it mean “using the adjusted R2 value to balance acceptable model fit with model parsimony”. I presumed that most of the regression model is based on likelihood. Maybe this approach is different, but I cannot understand the approach.

Author Response 4: 

Based on a significant rewrite of Section 2.3, we now explicitly state that the npsi parameter represents the number of permitted breakpoints in the “segmented” package. We hope this is clearer. Also, we feel that our complete reworking of this section provides more clarity (and simplicity) for the reader i.e., we have specifically pointed out for the reader that “perfect fit” (represented by R2 = 1) is essentially the time-series itself (e.g., total number of days in the time-series minus 1) . . . .. our approach essentially increases the npsi iteratively from 1 to 10 (in the first instance) to identify an npsi common across all sub-sets which is 1. Low enough for us to indicate specific individual dates/times for discussion, and 2. Represented by a appropriately high R2 whereby we can say with some certainty that we have successfully explained a high percentage of variance within the time-series/system. Our rewrite is included below. We believe it is important to point out that the provided reference (Muggeo, 2013) is the primary reference (and author of the “segmented” package), with all explanations of the approach explained from first principles here. We have simply tried to provide the reader with enough detail, rationale and the primary reference to ensure they can 1. Understand our study and 2. Replicate our approach if they wish. With respect to model parsimony, as an example, as shown in Table 3, npsi = 5 resulted in an adjusted R2 = 0.925, while npsi = 10 resulted in an adjusted R2 = 0.934 . . . . thus an extra 5 breakpoints does not provide a significantly improved overall time-series fit, and makes it far more difficult (for us and readers) to clearly interpret the resulting breakpoint model. We hope this makes sense. 

2.3 Analytical Methods 

“Segmented modelling via breakpoint regression is useful for assessing the effect of a covariate x (e.g., time-specific intervention) on the response y (e.g., incidence rate of infection), and has been widely used in medical and related research including mortality time-series [16], cancer incidence [17], and medication usage [18]. The “segmented” package [19] was used in the current study to fit several linear regression models between the response (laboratory-confirmed COVID-19 case number) and the explanatory variable (day number), thus allowing for identification of break-point estimates (i.e., dates on which COVID-19 incidence significantly increased/decreased). Identified breakpoints were subsequently compared with NPI phase dates using the World Health Organization’s (WHO) mean (5-day) and maximum (14-day) estimated COVID-19 incubation periods, in order to ascertain the likely direct effect of specific NPI phase changes with notable shifts in the COVID-19 incidence time-series (i.e., if breakpoints were identified within 5 to 14 days following an NPI phase date, the authors believe this likely indicates a relatively direct cause-effect relationship). In the current study, multiple breakpoints were permitted for identification (based on multiple NPI phases and observed COVID-19 trends in the Republic of Ireland during the study period), with the breakpoint linear regression model thus defined as: 

 yt = β0 + β1xt + δ1 (xt − τ1 ) + + ⋯ + δk (xt − τk ) +, t = 1,2, … , n (Eqn. 1)

where yt is daily infection incidence (i.e. number of confirmed infections) modelled as a linear function of the explanatory variable, xt, which is an ordinal number designating a day between 1 and 276 for the full time series, with τs (s = 1,2, . . , k) representing identified breakpoints. In this case, k breakpoints divide the time into (k + 1) intervals (or sections), with β representing calculated interval gradients i.e., β1 (first slope, xt < τ1 ), β2 = β1 + δ1 (second slope, i.e., first slope plus difference in slopes), etc. An iterative approach was taken to determine the optimum number of breakpoints, whereby the npsi parameter (i.e., number of breakpoints) was increased from 1 up to a maximum of 10, using the adjusted R2 value to optimise model fit without compromising model parsimony, as described by Muggeo (2013). In summary, for the current study a “perfect fit” would be provided by (x276 − 1) (i.e., 1 breakpoint per day within the time series, equating to the original time series). Thus, an increasing iterative approach was used to identify the lowest number of breakpoints required to explain the highest proportion of time-series variance and permit comparisons across sub-populations, without identifying identification of insignificant breakpoints (i.e., low gradient intervals).” 

Reviewer Comment 5. 

Line 188, xt is “day number”. Please describe what exact xt is in terms of the outbreak. If this only refers to symptomatic cases, why in Line 203, it was described “the entire dataset (N = 72,311, i.e., both symptomatic and asymptomatic) was included for breakpoint analyses…)”.

Author Response 5:

xt is an ordinal number designating individual days from 1 to 276 across the full time series. We have included this in our rewrite of Section 2.3 

We agree that our use of two sample numbers may be rather confusing for readers. Accordingly, we have sought to clarify this by amending the first results paragraph (below). We hope this is clearer.

“For individual (i.e., case-by-case) breakpoint analyses, 47,928 symptomatic COVID-19 cases (65.9% of all notified cases; 25,651 female (53.5%); mean age 41.2 years) were included, of which 61.5% (n = 29,459) of cases were classified as primary, and 18,469 (38.5%) classified as secondary cases. For breakpoint analyses of cluster number per day, the entire dataset including both symptomatic and asymptomatic cases (N = 72,311) was employed, based on the first reported epi-date associated with each cluster.”

Reviewer Comment 6. 

I don’t understand why look breakpoints of primary and secondary cases separately as shown in Figure 2. Why not look the sum of primary and secondary cases separately?

Author Response 6: 

As outlined in the discussion, we know that long-term care facilities had been severely impacted during the first wave, adding a large proportion of cases to overarching infection time-series, both in Ireland and internationally, and thus reasoned that these outbreaks/clusters could hardly be responsive to NPIs such as travel restrictions, implemented within the general population. This gave us an a priori reason to hypothesise that analysis of the primary and secondary cases would demonstrate different breakpoints. Our findings support this hypothesis, since the first breakpoint for secondary cases during the first wave occurs more than 20 days later than the breakpoint for primary cases. 

Reviewer Comment 7: 

The description in Table 1 and 2 are unclear. For example, why on Aug 18, Restaurants and Cafes to close by 11:30pm with a maximum of 6 per group (no more than 3 households) represent a decrease in restriction? Comparing to which one? Because on June 29 Cafes, restaurants, hotels, hostels, galleries, museums and pubs that serve food are allowed to open but social distancing must be maintained.

Author Response 7: 

In terms of increases/decreases in the stringency of restrictions, all increases and decreases are based on the period immediately prior to the new or adjusted restriction. We have added a footnote to Table 1 to indicate this. No mention of increased/decreased stringency is made with respect to Table 2. Table 2 presents the COVID-19 NPI Phases as set out by the Government of Ireland, as explicitly stated in the associated text 

Reviewer Comment 8. 

In the current draft, authors is not describing well the efficacy and lag period associated with specific NPIs. Where should we find the efficacy and the lag? Should the change in slope be interpreted as the efficacy? Why not estimate the growth rate in each interval? I will suggest to list all the effects of NPIs in the column of Table 2?

Author Response 8:

Thank you for your comment! The breakpoints indicate efficacy in a binary qualitative, “yes/no” sense. Although change in slope might be regarded as an indicator of efficacy, we have the following reservations: First, because multiple interventions were introduced concurrently, bundled together and implemented concurrently, it is very difficult to distinguish the effect of individual NPIs. Second, change in slope may be susceptible to the influence of factors such as population immunity and seasonality. Thus, we must express reservations about the use of change in slope a direct measure of efficacy. With respect to time lagged responses, we have alluded to these throughout the manuscript explaining to the reader that the primary indicator of a “direct response” to an NPI is identification of a significant breakpoint from 5-14 days post-NPI initiation, based on the WHO range of COVID-19 incubation. We hope this provides clarity!

We consider that interval slopes offer a measure of growth rate that may serve as a tentative indicator of intervention efficacy. However, the data are inherently limited in their potential to reveal the effects of individual NPIs, because the interventions were bundled together and implemented concurrently. For example, on 12 March, schools were closed, indoor and outdoor gatherings were banned, and workers were urged to work from home, while on March 15th, restaurants and public bars were closed. Accordingly, it is not possible to identify the effects of individual NPIs. We can at best seek to identify abrupt changes in incidence and work retrospectively, tentatively, to identify which “bundle” of NPIs (i.e., overarching phases) may have had an effect. Conversely, we can in some instances infer that particular NPI phases did not offer a significant, discernable effect, because there was no decrease/increase in slope within an interval of 14 days from their implementation, and such observations may be useful in guiding the future management of the epidemic. We have sought throughout to be extremely mindful and transparent as to the dataset and analytical limitations. 

Reviewer Comment 9: 

My suggestion is, since the breakpoint can be located different than the time NPI was implemented, the benefit of using these breakpoint should be clearly stated. In seems like breakpoints can be associated with deprivation score (In Line 318, ‘breakpoints identified within the sub-population associated with the “above-median”’). This may be a good reason. However, I am not so clear why these two are associated. Break points represent different time points. Please explain clearly how the association was determined.

Author Response 9: 

The authors would like to thank the reviewer for their suggestion. As stated, and not restated several times (for clarity) throughout the manuscript, identified breakpoints suggest abrupt changes in daily incidence (in addition to the date associated with this change), so they serve to retrospectively indicate which NPI phases were likely (in)effective based on a comparison of the breakpoint dates with NPI phase dates. Specifically, with regard to those NPIs implemented shortly (≤14 days based on WHO maximum COVID-19 incubation period) before an abrupt decrease in slope as most likely to have been effective, while those that were not followed by a decrease in slope within 14 days may be considered unlikely to have been effective. Accordingly, we have added the following text to the end of the methods section:

“The authors hypothesize that identified breakpoints suggest abrupt changes in daily incidence (in addition to the date associated with this change), thus serving to retrospectively indicate which NPI phases were likely (in)effective based on a comparison of the breakpoint dates with NPI phase dates. Specifically, regarding those NPIs implemented shortly (≤14 days based on WHO maximum COVID-19 incubation period) before an abrupt decrease in slope as likely to have been effective, while those that were not followed by a decrease in slope within 14 days may be considered unlikely to have been effective.”

Figure 6(c) indicates that the first three identified breakpoints relating to the “above median” deprivation sub-population (i.e., a more affluent sub-population) all occurred approximately 7-14 days prior to the equivalent breakpoint among the “below-median” subpopulation. Breakpoints within both sub-populations were identified/modelled in exactly the same way as all breakpoint modelling undertaken and presented within the manuscript. As such, we can (and have) said in the discussion that based on our analyses/findings, it would appear that more affluent populations responded more quickly to NPIs than their less affluent counterparts (See below). We hope this provides some clarity.

“The observed earlier first breakpoint among cases associated with a higher deprivation score (i.e., more affluent subset) may indicate a greater awareness and willingness to adopt voluntary measures for self-protection, while working from home (and thus complying with stay-at-home orders) may be more difficult for those in lower paid jobs. Conversely, the earlier timing of second and third breakpoints may point to higher levels of susceptibility among more deprived sub-populations, based on underlying health status, as mentioned above. A recently published retrospective cohort study comprising 3528 patients with laboratory confirmed COVID-19 residing in New York (US) found that patients associated with high poverty areas were significantly younger, had a higher prevalence of comorbidities and were more likely to be female or from a racial minority compared to individuals living in low poverty areas [37].”

Reviewer Comment 10. 

The discussion is too long, with 4 pages. Please make them concise to contain most important messages.

Author Response 10: 

We have further polished the discussion to focus on the most important findings, and it is now approximately 1600 words. We feel that due to the novelty of the work and outcomes, and the fact that it is the first study to emanate from Ireland, the current discussion at 1600 words, considering the volume of analyses presented, is now appropriate.

---

## [Editor Report · Decision Letter 1]

13 Jul 2021

Breakpoint modelling of temporal associations between non-pharmaceutical interventions and the incidence of symptomatic COVID-19 in the Republic of Ireland

PONE-D-21-10957R1

Dear Dr. Hynds,

We’re pleased to inform you that your manuscript has been judged scientifically suitable for publication and will be formally accepted for publication once it meets all outstanding technical requirements.

Kind regards,

Lucy C. Okell

Academic Editor

PLOS ONE
---

## [Editor Report · Acceptance letter]

15 Jul 2021

PONE-D-21-10957R1 

Breakpoint modelling of temporal associations between non-pharmaceutical interventions and symptomatic COVID-19 incidence in the Republic of Ireland 

Dear Dr. Hynds:

I'm pleased to inform you that your manuscript has been deemed suitable for publication in PLOS ONE. Congratulations! Your manuscript is now with our production department. 

Kind regards, 

on behalf of

Dr. Lucy C. Okell 

Academic Editor

PLOS ONE